# Neural Bayes: A Generic Parameterization Method for Unsupervised Learning

## Abstract

We introduce a parameterization method called Neural Bayes which allows computing statistical quantities that are in general difficult to compute and opens avenues for formulating new objectives for unsupervised representation learning. Specifically, given an observed random variable $\mathbf{x}$ and a latent discrete variable $z$, we can express $p(\mathbf{x}|z)$, $p(z|\mathbf{x})$ and $p(z)$ in closed form in terms of a sufficiently expressive function (Eg. neural network) using our parameterization without restricting the class of these distributions. To demonstrate its usefulness, we develop two independent use cases for this parameterization:

1. Disjoint Manifold Separation: Neural Bayes allows us to formulate an objective which can optimally label samples from disjoint manifolds present in the support of a continuous distribution. This can be seen as a specific form of clustering where each disjoint manifold in the support is a separate cluster. We design clustering tasks that obey this formulation and empirically show that the model optimally labels the disjoint manifolds.

2. Mutual Information Maximization (MIM): MIM has become a popular means for self-supervised representation learning. Neural Bayes allows us to compute mutual information between observed random variables $\mathbf{x}$ and latent discrete random variables $z$ in closed form. We use this for learning image representations and show its usefulness on downstream classification tasks.

## 1 Introduction

We introduce a generic parameterization called *Neural Bayes* that facilitates unsupervised learning from unlabeled data by categorizing them. Specifically, our parameterization implicitly maps samples from an observed random variable $\mathbf{x}$ to a latent discrete space $z$ where the distribution $p(\mathbf{x})$ gets segmented into a finite number of arbitrary conditional distributions. Imposing different conditions on the latent space $z$ through different objective functions will result in learning qualitatively different representations.

Our parameterization may be used to compute statistical quantities involving observed variables and latent variables that are in general difficult to compute (thanks to the discrete latent space), thus providing a flexible framework for unsupervised learning. To illustrate this aspect, we develop two independent use cases for this parameterization– disjoint manifold separation (DMS) and mutual information maximization (Linsker, 1988), as described in the abstract. For the manifold separation task, we show experiments on 2D datasets and their high-dimensional counter-parts designed as per the problem formulation, and show that the proposed objective can optimally label disjoint manifolds. For the MIM task, we experiment with benchmark image datasets and show that the unsupervised representation learned by the network achieves performance on downstream classification tasks comparable with a closely related MIM method Deep InfoMax (DIM, (Hjelm et al., 2019)). For both objectives we design regularizations necessary to achieve the desired behavior in practice. All the proofs can be found in the appendix.

## 2 Related Work

**Neural Bayes-DMS**: Numerous recent papers have proposed clustering algorithm for unsupervised representation learning such as Deep Clustering (Caron et al., 2018), information based clustering (Ji et al., 2019), Spectral Clustering (Shaham et al., 2018), Assosiative Deep Clustering (Haeusser

et al., 2018) etc. Our goal in regards to clustering in Neural Bayes-DMS is in general different from such methods. Our objective is aimed at finding disjoint manifolds in the support of a distribution. It is therefore a generalization of traditional subspace clustering methods (where the goal is to find disjoint affine subspaces) (Ma et al., 2008; Liu et al., 2010), to arbitrary manifolds.

Another class of clustering algorithms include mixture models (such as kNNs). Our clustering proposal (DMS) is novel compared to this class in two ways: 1. we formulate the clustering problem as that of identifying disjoint manifold in the support of a distribution. This is different from assuming K ground truth clusters, where the notion of cluster is ill-defined; 2. the DMS objective in proposition 1 is itself novel and we prove its optimality towards labeling disjoint manifolds in theorem 1.

**Neural Bayes-MIM**: Self-supervised representation learning has attracted a lot of attention in recent years. Currently contrastive learning methods and similar variants (such as MoCo (He et al., 2020), SimCLR (Chen et al., 2020), BYOL Grill et al. (2020)) produce state-of-the-art (SOTA) performance on downstream classification tasks. These methods make use of handcrafted image augmentation methods that exploit priors such as class information is typically associated with object shape and is location invariant. However, since we specifically develop an easier alternative to DIM (which also maximizes the mutual information between the input and the latent representations), for a fair comparison, we compare the performance of our Neural Bayes-MIM algorithm for representation learning only with DIM. We leave the extension of Neural Bayes MIM algorithm with data augmentation techniques and other advanced regularizations similar to Bachman et al. (2019) as future work. Our experiments show that our proposal performs comparably or slightly better compared with DIM. The main advantage of our proposal over DIM is that it offers a closed form estimation of MI due to discrete latent variables.

We note that the principle of mutual information maximization for representation learning was introduced in Linsker (1988) and Bell & Sejnowski (1995), and since then, a number of self-supervised methods involving MIM have been proposed. Vincent et al. (2010) showed that auto-encoder based methods achieve this goal implicitly by minimizing the reconstruction error of the input samples under isotropic Gaussian assumption. Deep infomax (DIM, Hjelm et al. (2019)) uses estimators like MINE Belghazi et al. (2018) and noise-contrastive estimation (NCE, Gutmann & Hyvärinen (2010)) to estimate MI and maximize it for both both local and global features in convolutional networks. Contrastive Predictive Coding (Oord et al., 2018) is another approach that maximizes MI by predicting the activations of a layer from the layer above using NCE.

We also point out that the estimation of mutual information due to Neural Bayes parameterization in the Neural Bayes-MIM-v1 objective (Eq 8) turns out to be identical to the one proposed in IMSAT (Hu et al., 2017). However, there are important differences: 1. we introduce regularizations which significantly improve the performance of representations on downstream tasks compared to IMSAT (cf table 3 in (Hu et al., 2017)); 2. we provide theoretical justifications for the parameterization used (lemma 1) and show in theorem 2 why it is feasible to compute high fidelity gradients using this objective in the mini-batch setting even though it contains the term $\mathbb{E}_{\mathbf{x}}[L_k(\mathbf{x})]$. On the other hand, the justification used in IMSAT is that optimizing using mini-batches is equivalent to optimizing an upper bound of the original objective; 3. we perform extensive ablation studies exposing the importance of the introduced regularizations.

## 3 NEURAL BAYES

Consider a data distribution $p(\mathbf{x})$ from which we have access to i.i.d. samples $\mathbf{x} \in \mathbb{R}^n$. We assume that this marginal distribution is a union of $K$ conditionals where the $k^{th}$ conditional's density is denoted by $p(\mathbf{x}|z = k) \in \mathbb{R}^+$ and the corresponding probability mass denoted by $p(z = k) \in \mathbb{R}^+$. Here $z$ is a discrete random variable with $K$ states. We now introduce the parameterization that allows us to implicitly factorize any marginal distribution into conditionals as described above. Aside from the technical details, the key idea behind this parameterization is the Bayes' rule.

**Lemma 1.** *Let $p(\mathbf{x}|z = k)$ and $p(z)$ be any conditional and marginal distribution defined for continuous random variable $\mathbf{x}$ and discrete random variable $z$. If $\mathbb{E}_{\mathbf{x} \sim p(\mathbf{x})}[L_k(\mathbf{x})] \neq 0 \ \forall k \in [K]$, then there exists a non-parametric function $L(\mathbf{x}) : \mathbb{R}^n \to \mathbb{R}^{+K}$ for any given input $\mathbf{x} \in \mathbb{R}^n$ with the*

property $\sum_{k=1}^{K} L_k(\mathbf{x}) = 1 \; \forall \mathbf{x}$ *such that,*

$$p(\mathbf{x}|z=k) = \frac{L_k(\mathbf{x}) \cdot p(\mathbf{x})}{\mathbb{E}_{\mathbf{x} \sim p(\mathbf{x})}[L_k(\mathbf{x})]}, \quad p(z=k) = \mathbb{E}_{\mathbf{x}}[L_k(\mathbf{x})], \quad p(z=k|\mathbf{x}) \quad = L_k(\mathbf{x}) \quad (1)$$

Thus the function $L$ can be seen as a form of soft categorization of input samples. In practice, we use a neural network with sufficient capacity and softmax output to realize this function $L$. We name our parameterization method *Neural Bayes* and replace $L$ with $L_\theta$ to denote the parameters of the network. By imposing different conditions on the structure of $z$ by formulating meaningful objectives, we will get qualitatively different kinds of factorization of the marginal $p(\mathbf{x})$, and therefore the function $L_\theta$ will encode the posterior for that factorization. In summary, if one formulates any objective that involves the terms $p(\mathbf{x}|z)$, $p(z)$ or $p(z|\mathbf{x})$, where $\mathbf{x}$ is an observed random variable and $z$ is a discrete latent random variable, then they can be substituted with $\frac{L_k(\mathbf{x}) \cdot p(\mathbf{x})}{\mathbb{E}_{\mathbf{x}}[L_k(\mathbf{x})]}$, $\mathbb{E}_{\mathbf{x}}[L_k(\mathbf{x})]$ and $L_k(\mathbf{x})$ respectively.

On an important note, Neural Bayes parameterization requires using the term $\mathbb{E}_{\mathbf{x}}[L_k(\mathbf{x})]$, through which computing gradient is infeasible in general. A general discussion around this can be found in the appendix D. Nonetheless, we show that mini-batch gradients can have good fidelity for one of the objectives we propose using our parameterization. In the next two sections, we explore two different ways of factorizing $p(\mathbf{x})$ resulting in qualitatively different goals of unsupervised representation learning.

## 4 DISJOINT MANIFOLD SEPARATION (DMS)

In many cases, the support of a distribution may be a set of disjoint manifolds. In this task, our goal is to label samples from each disjoint manifold with a distinct value. This formulation can be seen as a generalization of subspace clustering (Ma et al., 2008) where the goal is to identify disjoint affine subspaces. To make the problem concrete, we first formalize the definition of a disjoint manifold.

**Definition 1.** *(Connected Set) We say that a set $S \subset \mathbb{R}^n$ is a connected set (disjoint manifold) if for any $\mathbf{x}, \mathbf{y} \in S$, there exists a continuous path between $\mathbf{x}$ and $\mathbf{y}$ such that all the points on the path also belong to $S$.*

To identify such disjoint manifolds in a distribution, we exploit the observation that only partitions that separate one disjoint manifold from others have high divergence between the respective conditional distributions while partitions that cut through a disjoint manifold result in conditional distributions with low divergence between them. Therefore, the objective we propose for this task is to partition the unlabeled data distribution $p(\mathbf{x})$ into conditional distributions $q_i(\mathbf{x})$'s such that a divergence between them is maximized. By doing so we recover the conditional distributions defined over the disjoint manifolds (we prove its optimality in theorem 1). We begin with two disjoint manifolds and extend this idea to multiple disjoint manifolds in the appendix B.

Let $J$ be a symmetric divergence (Eg. Jensen-Shannon divergence, Wasserstein divergence, etc), and $q_0$ and $q_1$ be the disjoint conditional distributions that we want to learn. Then the aforementioned objective can be written formally as follows:

$$\max_{\substack{q_0, q_1 \\ \pi \in (0,1)}} J(q_0(\mathbf{x})||q_1(\mathbf{x})) \tag{2}$$

$$\text{s.t.} \quad \int_{\mathbf{x}} q_0(\mathbf{x}) = 1, \quad \int_{\mathbf{x}} q_1(\mathbf{x}) = 1, \quad q_1(\mathbf{x}) \cdot \pi + q_0(\mathbf{x}) \cdot (1-\pi) = p(\mathbf{x}).$$

Since our goal is to simply assign labels to data samples $\mathbf{x}$ corresponding to which manifold they belong instead of learning conditional distributions as achieved by Eq. (2), we would like to learn a function $L(\mathbf{x})$ which maps samples from disjoint manifolds to distinct labels. To do so, below we derive an objective equivalent to Eq. (2) that learns such a function $L(\mathbf{x})$.

**Proposition 1.** *(Neural Bayes-DMS) Let $L(\mathbf{x}) : \mathbb{R}^n \to [0, 1]$ be a non-parametric function for any given input $\mathbf{x} \in \mathbb{R}^n$, and let $J$ be the Jensen-Shannon divergence. Define scalars $f_1(\mathbf{x}) := \frac{L(\mathbf{x})}{\mathbb{E}_{\mathbf{x}}[L(\mathbf{x})]}$*

and $f_0(\mathbf{x}) := \frac{1 - L(\mathbf{x})}{1 - \mathbb{E}_{\mathbf{x}}[L(\mathbf{x})]}$. *Then the objective in Eq. (2) is equivalent to,*

$$\max_L \frac{1}{2} \cdot \mathbb{E}_{\mathbf{x}} \left[ f_1(\mathbf{x}) \cdot \log \left( \frac{f_1(\mathbf{x})}{f_1(\mathbf{x}) + f_0(\mathbf{x})} \right) \right] + \frac{1}{2} \cdot \mathbb{E}_{\mathbf{x}} \left[ f_0(\mathbf{x}) \cdot \log \left( \frac{f_0(\mathbf{x})}{f_1(\mathbf{x}) + f_0(\mathbf{x})} \right) \right] + \log 2$$
$$s.t. \quad \mathbb{E}_{\mathbf{x}}[L(\mathbf{x})] \notin \{0, 1\}. \tag{3}$$

**Optimality**: We now prove the optimality of the proposed objective towards discovering disjoint manifolds present in the support of a probability density function $p(\mathbf{x})$.

**Theorem 1.** *(optimality) Let $p(\mathbf{x})$ be a probability density function over $\mathbb{R}^n$ whose support is the union of two non-empty connected sets (definition 1) $S_1$ and $S_2$ that are disjoint, i.e. $S_1 \cap S_2 = \varnothing$. Let $L(\mathbf{x}) \in [0, 1]$ belong to the class of continuous functions which is learned by solving the objective in Eq. (3). Then the objective in Eq. (3) is maximized if and only if one of the following is true:*

$$L(\mathbf{x}) = \begin{cases} 0 & \forall \mathbf{x} \in S_1 \\ 1 & \forall \mathbf{x} \in S_2 \end{cases} \quad or \quad L(\mathbf{x}) = \begin{cases} 1 & \forall \mathbf{x} \in S_1 \\ 0 & \forall \mathbf{x} \in S_2. \end{cases}$$

The above theorem proves that optimizing the derived objective over the space of functions $L$ implicitly partitions the data distribution into maximally separated conditionals by assigning a distinct label to points in each manifold. Most importantly, the theorem shows that the continuity condition on the function $L(\mathbf{x})$ plays an important role. Without this condition, the network cannot identify disjoint manifolds. Extension to multiple disjoint manifold case can be found in section B in appendix.

## 4.1 IMPLEMENTATION DETAILS

**Prior Collapse**: The constraint in proposition 1 is a boundary condition required for technical reasons in lemma 1. In practice we do not worry about them because optimization itself avoids situations where $\mathbb{E}_{\mathbf{x}}[L(\mathbf{x})] \in \{0, 1\}$. To see the reason behind this, note that except when initialized in a way such that $\mathbb{E}_{\mathbf{x}}[L(\mathbf{x})] \in \{0, 1\}$, the log terms are negative by definition. Since the denominators of $f_0$ and $f_1$ are $\mathbb{E}_{\mathbf{x}}[L(\mathbf{x})]$ and $1 - \mathbb{E}_{\mathbf{x}}[L(\mathbf{x})]$ respectively, the objective is maximized when $\mathbb{E}_{\mathbf{x}}[L(\mathbf{x})]$ moves away from 0 and 1. Thus, for any reasonable initialization, optimization itself pushes $\mathbb{E}_{\mathbf{x}}[L(\mathbf{x})]$ away from 0 and 1.

**Smoothness** of $L_\theta(.)$: As shown in theorem 1, the proposed objectives can optimally recover disjoint manifolds only when the function $L_\theta(.)$ is continuous. In practice we found enforcing the function to be smooth (thus also continuous) helps significantly. Therefore, after experimenting with a handful of heuristics for regularizing $L_\theta$, we found the following finite difference Jacobian regularization to be effective ($L(.)$ can be scalar or vector),

$$\mathcal{R}_c = \frac{1}{B} \sum_{i=1}^{B} \frac{\|L_\theta(\mathbf{x}_i) - L_\theta(\mathbf{x}_i + \zeta \cdot \hat{\delta}_i)\|^2}{\zeta^2} \tag{4}$$

where $\hat{\delta}_i := \frac{\delta_i}{\|\delta_i\|_2}$ is a normalized noise vector computed independently for each sample $\mathbf{x}_i$ in a batch of size $B$ as $\delta_i := \mathbf{X} \mathbf{v}_i$. Here $\mathbf{X} \in \mathbb{R}^{n \times B}$ is the matrix containing the batch of samples, and each dimension of $\mathbf{v}_i \in \mathbb{R}^B$ is sampled i.i.d. from a standard Gaussian. This computation ensures that the perturbation lies in the span of data, which we found to be important. Finally $\zeta$ is the scale of normalized noise added to all samples in a batch. In our experiments, since we always normalize the datasets to have zero mean and unit variance across all dimensions, we sample $\zeta \sim \mathcal{N}(0, 0.1^2)$.

**Implementation**: We implement the binary-partition Neural Bayes-DMS using the Monte-Carlo sampling approximation of the following objective,

$$\min_\theta \frac{1}{2} \cdot \mathbb{E}_{\mathbf{x}} \left[ f_1(\mathbf{x}) \cdot \log \left( 1 + \frac{f_0(\mathbf{x})}{f_1(\mathbf{x})} \right) \right] + \frac{1}{2} \cdot \mathbb{E}_{\mathbf{x}} \left[ f_0(\mathbf{x}) \cdot \log \left( 1 + \frac{f_1(\mathbf{x})}{f_0(\mathbf{x})} \right) \right] + \beta \cdot \mathcal{R}_c \tag{5}$$

where $f_1(\mathbf{x}) := \frac{L_\theta(\mathbf{x})}{\mathbb{E}_{\mathbf{x}}[L_\theta(\mathbf{x})]} + \epsilon$ and $f_0(\mathbf{x}) := \frac{1 - L_\theta(\mathbf{x})}{1 - \mathbb{E}_{\mathbf{x}}[L_\theta(\mathbf{x})]} + \epsilon$. Here $\epsilon = 10^{-7}$ is a small scalar used to prevent numerical instability, and $\beta$ is a hyper-parameter to control the continuity of $L$. The multi-partition case can be implemented in a similar way. Due to the need for computing $\mathbb{E}_{\mathbf{x}}[L_\theta(\mathbf{x})]$ in the objective, optimizing it using gradient descent methods with small batch-sizes is not possible. Therefore we experiment with this method on datasets where gradients can be computed for a very large batch-size needed to approximate the gradient through $\mathbb{E}_{\mathbf{x}}[L_\theta(\mathbf{x})]$ sufficiently well.

## 4.2 EXPERIMENTS

Clustering in general is an ill posed problem. However, in our problem setup, the definition is precise, i.e., our goal is to optimally label all the disjoint manifolds present in the support of a distribution. Since this is a unique goal that is not generally considered in literature, as empirical verification, we show qualitative results on 2D synthetic datasets in figure 1. Top 2 sub-figures have 2 clusters and the bottom 2 have 3 clusters. For all experiments we use a 4 layer MLP with 400 hidden units each, batchnorm, ReLU activation, and last layer Softmax activation. In all cases we train using Adam optimizer with a learning rate of 0.001, batch size of 400 and no weight decay, and trained until convergence. Regularization coefficient $\beta$ was chosen from $[0.5, 6]$ that resulted in optimal clustering. For generality in these experiments, these 2D datasets were projected to high dimensions (512) by appending 510 dimensions of 0 entries to each sample and then randomly rotated before performing clustering. The datasets were then projected back to the original 2D space for visualizing predictions. Additional experiments can be found in the appendix C.

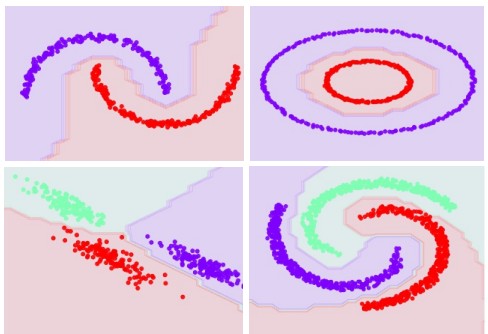

Figure 1: Neural Bayes-DMS network prediction on synthetic high dimensional data (lying on a 2D plane) projected back to 2 dimensions for visualization. The different colors denote the label predicted by the network $L_\theta(.)$ thresholded at 0.5. The darker shades of colors denote predictions made at training data points while the lighter shades denote predictions on the rest of the space.

## 5 MUTUAL INFORMATION MAXIMIZATION (MIM)

Suppose we want to find a discrete latent representation $z$ (with $K$ states) for the distribution $p(\mathbf{x})$ such that the mutual information $MI(\mathbf{x}, z)$ is maximized (Linsker, 1988). Such an encoding $z$ demands that it must be very efficient since it has to capture maximum possible information about the continuous distribution $p(\mathbf{x})$ in just $K$ discrete states. Assuming we can learn such an encoding, we are interested in computing $p(z|\mathbf{x})$ since it tells us the likelihood of $\mathbf{x}$ belonging to each discrete state of $z$, thereby performing soft categorization which may be useful for downstream tasks. In the proposition below, we show an objective for computing $p(z|\mathbf{x})$ for a discrete latent representation $z$ that maximizes $MI(\mathbf{x}, z)$.

**Proposition 2.** *(Neural Bayes-MIM-v1) Let $L(\mathbf{x}) : \mathbb{R}^n \to \mathbb{R}^{+^K}$ be a non-parametric function for any given input $\mathbf{x} \in \mathbb{R}^n$ with the property $\sum_{i=1}^K L_k(\mathbf{x}) = 1 \ \forall \mathbf{x}$. Consider the following objective,*

$$L^* = \arg\max_L \mathbb{E}_{\mathbf{x}} \left[ \sum_{k=1}^K L_k(\mathbf{x}) \log \frac{L_k(\mathbf{x})}{\mathbb{E}_{\mathbf{x}}[L_k(\mathbf{x})]} \right] \tag{6}$$

*Then $L_k^*(\mathbf{x}) = p(z^* = k|\mathbf{x})$, where $z^* \in \arg\max_z MI(\mathbf{x}, z)$.*

The proof essentially involves expressing MI in terms of $p(z|\mathbf{x})$, and $p(z)$, which can be substituted using Neural Bayes parameterization. However, the objective proposed in the above theorem poses a challenge– the objective contains the term $\mathbb{E}_{\mathbf{x}}[L_k(\mathbf{x})]$ for which computing high fidelity gradient in a batch setting is problematic (see appendix D). However, we can overcome this problem for the MIM objective because it turns out that gradient through certain terms are 0 as shown by the following theorem.

**Theorem 2.** *(Gradient Simplification) Denote,*

$$J(\theta) = -\mathbb{E}_{\mathbf{x}} \left[ \sum_{k=1}^K L_{\theta_k}(\mathbf{x}) \log \frac{L_{\theta_k}(\mathbf{x})}{\mathbb{E}_{\mathbf{x}}[L_{\theta_k}(\mathbf{x})]} \right], \quad \hat{J}(\theta) = -\mathbb{E}_{\mathbf{x}} \left[ \sum_{k=1}^K L_{\theta_k}(\mathbf{x}) \log \langle \frac{L_{\theta_k}(\mathbf{x})}{\mathbb{E}_{\mathbf{x}}[L_{\theta_k}(\mathbf{x})]} \rangle \right] \tag{7}$$

*where $\langle . \rangle$ indicates that gradients are not computed through the argument. Then $\frac{\partial J(\theta)}{\partial \theta} = \frac{\partial \hat{J}(\theta)}{\partial \theta}$.*

The above theorem implies that as long as we plugin a decent estimate of $\mathbb{E}_{\mathbf{x}}[L_{\theta_k}(\mathbf{x})]$ in the objective, unbiased gradients can be computed using randomly sampled mini-batches. Note that the objective can be re-written as,

$$\min_\theta - \mathbb{E}_{\mathbf{x}} \left[ \sum_{k=1}^K L_{\theta_k}(\mathbf{x}) \log\langle L_{\theta_k}(\mathbf{x}) \rangle \right] + \sum_{k=1}^K \mathbb{E}_{\mathbf{x}}[L_k(\mathbf{x})] \log\langle \mathbb{E}_{\mathbf{x}}[L_k(\mathbf{x})] \rangle \tag{8}$$

The second term is the negative entropy of the discrete latent representation $p(z = k) := \mathbb{E}_{\mathbf{x}}[L_k(\mathbf{x})]$ which acts as a uniform prior. In other words, this term encourages learning a latent code $z$ such that all states of $z$ activate uniformly over the marginal input distribution $\mathbf{x}$. This is an attribute of distributed representation which is a fundamental goal in deep learning. We can therefore further encourage this behavior by treating the coefficient of this term as a hyper-parameter. In our experiments we confirm both the distributed representation behavior of this term as well as the benefit of using a hyper-parameter as our coefficient.

### 5.1 IMPLEMENTATION DETAILS

**Alternative Formulation of Uniform Prior**: In practice we found that an alternative formulation of the second term in Eq 8 results in better performance and more interpretable filters. Specifically, we replace it with the following cross-entropy formulation,

$$\mathcal{R}_p(\theta) := - \sum_{k=1}^K \frac{1}{K} \log(\mathbb{E}_{\mathbf{x}}[L_k(\mathbf{x})]) + \frac{K-1}{K} \log(1 - \mathbb{E}_{\mathbf{x}}[L_k(\mathbf{x})]) \tag{9}$$

While both, the second term in Eq 8 as well as $\mathcal{R}_p(\theta)$ are minimized when $\mathbb{E}_{\mathbf{x}}[L_k(\mathbf{x})] = 1/K$, the latter formulation provides much stronger gradients during optimization when $\mathbb{E}_{\mathbf{x}}[L_k(\mathbf{x})]$ approaches 1 (see appendix F.1 for details); $\mathbb{E}_{\mathbf{x}}[L_k(\mathbf{x})] = 1$ is undesirable since it discourages distributed representation. Finally, unbiased gradients can be computed through Eq 9 as long as a good estimate of $\mathbb{E}_{\mathbf{x}}[L_k(\mathbf{x})]$ is plugged in. Also note that the condition $\mathbb{E}_{\mathbf{x}}[L_k(\mathbf{x})] \notin \{0, 1\}$ in lemma 1 is met by the Neural Bayes-MIM objective implicitly during optimization as discussed in the above paragraph in regards to distributed representation.

**Implementation**: The final Neural Bayes-MIM-v2 objective is,

$$\min_\theta - \mathbb{E}_{\mathbf{x}} \left[ \sum_{k=1}^K L_{\theta_k}(\mathbf{x}) \log\langle L_{\theta_k}(\mathbf{x}) + \epsilon \rangle \right] + (1 + \alpha) \cdot \mathcal{R}_p(\theta) + \beta \cdot \mathcal{R}_c \tag{10}$$

where $\alpha$ and $\beta$ are hyper-parameters, $\mathcal{R}_c$ is a smoothness regularization introduced in Eq. 4, $\epsilon = 10^{-7}$ is a small scalar used to prevent numerical instability. Qualitatively, we find that the regularization $\mathcal{R}_c$ prevents filters from memorizing the input samples. Finally, we apply the first two terms in Eq 10 to all hidden layers of a deep network at different scales (computed by spatially average pooling and applying Softmax). These two regularizations gave a significant performance boost. Thorough implementation details are provided in the appendix E. For brevity, we refer to our final objective as Neural Bayes-MIM in the rest of the paper.

On the other hand, to compute a good estimate of gradients, we use the following trick. During optimization, we compute gradients using a sufficiently large mini-batch of size MBS (Eg. 500) that fits in memory (so that the estimate of $\mathbb{E}_{\mathbf{x}}[L_k(\mathbf{x})]$ is reasonable), and accumulate these gradients until BS samples are seen (Eg. 2000), and averaged before updating the parameters to further reduce estimation error.

### 5.2 EXPERIMENTS

Instead of aiming for state-of-the-art results on self-supervised learning tasks, our goal in this section is to conduct experiments using Neural Bayes-MIM to understand the behavior of the algorithm, the hyper-parameters involved, and compare performance with DIM, a closely related self-supervised learning method. The use of data augmentation, additional architectures and other regularizations similar to Bachman et al. (2019) is left as future work. Therefore, we use the following simple CNN encoder architecture[1] $Enc$ in our experiments: $C(200, 3, 1, 0) - P(2, 2, 0, \max) - C(500, 3, 1, 0) -$

---

[1] We use the following shorthand for a) conv layer: C(number of filters, filter size, stride size, padding); b) pooling: P(kernel size, stride, padding, pool mode)

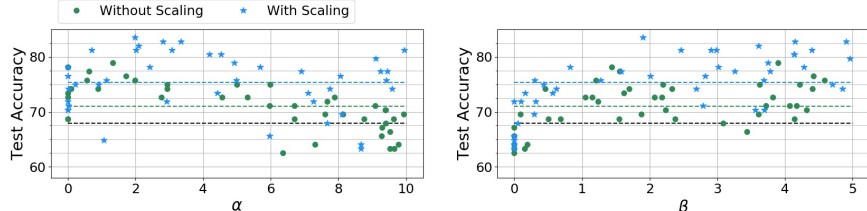

Figure 2: The effect of Neural Bayes-MIM hyper-parameters on the final test performance for CIFAR-10. A CNN encoder is trained using Neural Bayes-MIM with different configurations–hyper-parameters $\alpha$, $\beta$ and scaling of states on which the objective is applied. A one hidden layer classifier (with 200 units) is then trained using labels on these frozen features to get the final test accuracy. The plots show that performance is significantly worse when $\alpha = \beta = 0$ and no scaling is used, showing their important role as a regularizer. The best performing model reaches 83.59%. Black dotted line is the baseline performance (67.97%) when a randomly initialized network (with identical architecture) is used as encoder. Green and blue dotted lines are the average of all the green and blue points respectively.

$C(700, 3, 1, 0) - P(2, 2, 0, \max) - C(1000, 3, 1, 0)$. For an input image $\mathbf{x}$ of size $32 \times 32 \times 3$, the output of this encoder $Enc(\mathbf{x})$ has size $2 \times 2 \times 1000$. The encoder is initialized using orthogonal initialization (Saxe et al., 2013), batch normalization (Ioffe & Szegedy, 2015) is used after each convolution layer and ReLU non-linearities are used. All datasets are normalized to have dimension-wise 0 mean and unit variance. Early stopping in all experiments is done using the test set (following previous work). We broadly follow the experimental setup of Hjelm et al. (2019). We do not use any data augmentation in our experiments. After training the encoder, we freeze its features and train a 1 hidden layer (200 units) classifier to get the final test accuracy.

### 5.2.1 ABLATION STUDIES

**Behavior of Neural Bayes-MIM-v1 (Eq 8) vs Neural Bayes-MIM (v2, Eq 9)**: The experiments and details are discussed in the appendix F.2. The main differences are: 1. majority of the filters learned by the v1 objective are dead, as opposed to the v2 objective which encourages distributed representation; 2. the performance of v2 is better than that of the v1 objective.

**Performance due to Regularizations and State Scaling**: We now evaluate the effects of the various components involved in the Neural Bayes-MIM objective– coefficients $\alpha$ and $\beta$, and applying the objective at different scales of hidden states. We use the CIFAR-10 dataset for these experiments.

In the first experiment, for each value of the number of different scales considered, we vary $\alpha$, $\beta$ and record the final performance, thus capturing the variation in performance due to all these three components. We consider two scaling configurations: 1. no pooling is applied to the hidden layers; 2. for each hidden layer, we spatially average pool the state using a $2 \times 2$ pooing filter with a stride of 2. For the encoder used in our experiments (which has 4 internal hidden layers post ReLU), this gives us 4 and 8 states respectively (including the original un-scaled hidden layers) to apply the Neural Bayes-MIM objective. After getting all the states, we apply the Softmax activation to each state along the channel dimension so that the Neural Bayes parameterization holds. Thus for states with height and width, the objective is applied to each spatial $(x, y)$ location separately and averaged. Also, for states with height (or width) less than the pooling size, we use the height (or width) as pooling size.

We train Neural Bayes-MIM on the full training set for 100 epochs using Adam with learning rate 0.001 (other Adam hyper-parameters are standard), mini-batch size 500 and batch size 2000, 0 weight decay. In the first 32 experiments, $\alpha$ and $\beta$ are sampled uniformly from $[0, 10]$ and $[0, 5]$ respectively. In the next 5 experiments, $\alpha$ is set to be 0 while $\beta$ is sampled uniformly. In the next 5 experiments, $\beta$ is set to be 0 while $\alpha$ is sampled uniformly. Thus in total we run 42 experiments for each number of scaling considered.

Once we get a trained $Enc(\mathbf{x})$, we train a 1 hidden layer (with 200 units) MLP classifier on the frozen features from $Enc(\mathbf{x})$ using the labels in the training set. This training is done for 100 epochs using Adam with learning rate 0.001 (other Adam hyper-parameters are standard), batch size 128 and weight decay 0.

| Encoder \Dataset | CIFAR-10 | CIFAR-100 | STL-10 |
|---|---|---|---|
| Random Network | 67.97 | 42.97 | 53.91 |
| DIM (Hjelm et al., 2019) | 80.95 | 49.74 | - |
| Neural Bayes-MIM | **82.81** | **55.47** | **64.84** |

Table 1: Classification performance of a one hidden layer MLP classifier trained on frozen features from the mentioned encoder models (trained using Neural Bayes-MIM) and datasets (no data augmentation used in experiments we ran). STL-10 was resized to $32 \times 32$ in our runs instead of $64 \times 64$ as in DIM due to memory restrictions. Performance reported from DIM paper are their best numbers (omitting STL-10 due to difference in image size).

As a baseline for these experiments, we use a randomly initialed encoder $Enc(\mathbf{x})$. Since there are no tunable hyper-parameters in this case, we perform a grid search on the classifier hyper-parameters. Specifically, we choose weight decay from $\{0, 1e-5, 5e-5, 1e-4\}$, batch size from $\{128, 256\}$, and learning rate from $\{0.0001, 0.001\}$. This yields a total of 16 configurations. The test accuracy from these runs varied between 58.59% and 67.97%. We consider 67.97% as our baseline.

The performance of encoders under the aforementioned configurations is shown in figure 2. It is clear that both the hyper-parameters $\alpha$ and especially $\beta$ play an important role in the quality of representations learned. Also, applying Neural Bayes-MIM at different scales of the network states significantly improves the average and best performance.

Filter visualization, convergence experiments and the effect of batch-size are shown in appendix F.3.

### 5.2.2 FINAL CLASSIFICATION PERFORMANCE

We compare the final test accuracy of Neural Bayes-MIM with 2 baselines– a random encoder (described in ablation studies) and Deep Infomax (Hjelm et al., 2019) on benchmark image datasets– CIFAR10 and CIFAR-100 (Krizhevsky, 2009) and STL-10 (Coates et al., 2011). Random Network refers to the use of a randomly initialized network. The experimental details for them are identical to those in the ablation above involving a hyper-parameter search over 16 configurations done for each dataset separately.

DIM results are reported from Hjelm et al. (2019). We omit STL-10 number for DIM because we resize images to a much smaller size of $32 \times 32$ in our runs instead of $64 \times 64$ as used in DIM.

The following describes the experimental details for Neural Bayes-MIM. We use $\alpha = 2$ and $\beta = 4$ (chosen roughly by examining figure 2), and MBS=500, BS=4000 in all the experiments. Note these values are not tuned for STL-10 and CIFAR-100. For CIFAR-10 and STL-10 each, we run 4 configurations of Neural Bayes-MIM over hyper-parameters learning rate $\in \{0.0001, 0.001\}$ and weight decay $\in \{0, 0.00001\}$. For each run, we then train a 1 hidden layer (200 units) classifier on top of the frozen features with learning rate $\in \{0.0001, 0.001\}$. We report the best performance of all runs. For CIFAR-100, we take the encoder that produces the best performance on CIFAR-10, and train a classifier with the 2 learning rates and report the best of the 2 runs.

Table 1 reports the classification performance of all the methods. We note that all experiments were done with CNN architecture without any data augmentation. Neural Bayes-MIM outperforms baseline methods in general.

## 6 CONCLUSION

We proposed a parameterization method that can be used to express an arbitrary set of distributions $p(\mathbf{x}|z)$, $p(z|\mathbf{x})$ and $p(z)$ in closed form using a neural network with sufficient capacity, which can in turn be used to formulate new objective functions. We formulated two different objectives that use this parameterization which were aimed towards different goals of unsupervised learning– identification of disjoint manifolds in the support of continuous distributions and learning deep network features using the infomax principle. We focused on theoretical analysis for both the objectives and presented preliminary experiments supporting the theoretical claims.

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

APPENDIX

## A PROOFS FOR NEURAL BAYES-DMS (BINARY CASE)

**Proposition 1.** *(Neural Bayes-DMS, proposition 2 in main text) Let $L(\mathbf{x}) : \mathbb{R}^n \to [0, 1]$ be a non-parametric function for any given input $\mathbf{x} \in \mathbb{R}^n$, and let $J$ be the Jensen-Shannon divergence. Define scalars $f_1(\mathbf{x}) := \frac{L(\mathbf{x})}{\mathbb{E}_\mathbf{x}[L(\mathbf{x})]}$ and $f_0(\mathbf{x}) := \frac{1-L(\mathbf{x})}{1-\mathbb{E}_\mathbf{x}[L(\mathbf{x})]}$. Then the objective in Eq. (2) is equivalent to,*

$$\max_L \frac{1}{2} \cdot \mathbb{E}_\mathbf{x} \left[ f_1(\mathbf{x}) \cdot \log \left( \frac{f_1(\mathbf{x})}{f_1(\mathbf{x}) + f_0(\mathbf{x})} \right) \right] + \frac{1}{2} \cdot \mathbb{E}_\mathbf{x} \left[ f_0(\mathbf{x}) \cdot \log \left( \frac{f_0(\mathbf{x})}{f_1(\mathbf{x}) + f_0(\mathbf{x})} \right) \right] + \log 2 \tag{11}$$

$$s.t. \quad \mathbb{E}_\mathbf{x}[L(\mathbf{x})] \notin \{0, 1\} \tag{12}$$

*Proof: Using the Neural Bayes parameterization from lemma 1 for binary case, we set,*

$$q_1(\mathbf{x}) := \frac{L(\mathbf{x}) \cdot p(\mathbf{x})}{\mathbb{E}_{\mathbf{x} \sim p(\mathbf{x})}[L(\mathbf{x})]} \quad q_0(\mathbf{x}) := \frac{(1 - L(\mathbf{x})) \cdot p(\mathbf{x})}{1 - \mathbb{E}_{\mathbf{x} \sim p(\mathbf{x})}[L(\mathbf{x})]} \tag{13}$$

*These parameterizations therefore automatically satisfy the constraints in Eq. (2). Finally, using the definition of JS divergence, the maximization problem in Eq. (2) can be written as,*

$$\max_{q_0, q_1} \frac{1}{2} \cdot \int_\mathbf{x} q_1(\mathbf{x}) \log \frac{q_1(\mathbf{x})}{0.5 \cdot (q_0(\mathbf{x}) + q_1(\mathbf{x}))} + q_0(\mathbf{x}) \log \frac{q_0(\mathbf{x})}{0.5 \cdot (q_0(\mathbf{x}) + q_1(\mathbf{x}))} \tag{14}$$

*Substituting $q_0$ and $q_1$ with their respective parameterizations and using the definitions of $f_0(\mathbf{x})$ and $f_1(\mathbf{x})$ completes the proof.* $\square$

**Theorem 1.** *(optimality, Theorem 2 in main text) Let $p(\mathbf{x})$ be a probability density function over $\mathbb{R}^n$ whose support is the union of two non-empty connected sets (definition 1) $S_1$ and $S_2$ that are disjoint, i.e. $S_1 \cap S_2 = \varnothing$. Let $L(\mathbf{x}) \in [0, 1]$ belong to the class of continuous functions which is learned by solving the objective in Eq. (3). Then the objective in Eq. (3) is maximized if and only if one of the following is true:*

$$L(\mathbf{x}) = \begin{cases} 0 & \forall \mathbf{x} \in S_1 \\ 1 & \forall \mathbf{x} \in S_2 \end{cases} \quad or \quad L(\mathbf{x}) = \begin{cases} 1 & \forall \mathbf{x} \in S_1 \\ 0 & \forall \mathbf{x} \in S_2 \end{cases} \tag{15}$$

*Proof: The two cases exist in the theorem due to symmetry. Recall the definition of $f_0(\mathbf{x})$ and $f_1(\mathbf{x})$ in Eq. (3),*

$$f_1(\mathbf{x}) := \frac{L(\mathbf{x})}{\mathbb{E}_\mathbf{x}[L(\mathbf{x})]} \qquad f_0(\mathbf{x}) := \frac{1 - L(\mathbf{x})}{1 - \mathbb{E}_\mathbf{x}[L(\mathbf{x})]} \tag{16}$$

*where $L(\mathbf{x}) \in [0, 1]$ and for a feasible $L(\mathbf{x})$, and therefore $\pi := \mathbb{E}_\mathbf{x}[L(\mathbf{x})] \in (0, 1)$ due to the conditions specified in this theorem. Thus $f_1(\mathbf{x}) \in [0, \frac{1}{\pi}]$ and $f_0(\mathbf{x}) \in [0, \frac{1}{1-\pi}]$. By design, the terms $\log \left( \frac{f_1(\mathbf{x})}{f_1(\mathbf{x})+f_0(\mathbf{x})} \right)$ and $\log \left( \frac{f_0(\mathbf{x})}{f_1(\mathbf{x})+f_0(\mathbf{x})} \right)$ are non-positive. Thus, for any $\mathbf{x} \in S_1 \cup S_2$,*

$$F(\mathbf{x}) = f_1(\mathbf{x}) \cdot \log \left( \frac{f_1(\mathbf{x})}{f_1(\mathbf{x}) + f_0(\mathbf{x})} \right) + f_0(\mathbf{x}) \cdot \log \left( \frac{f_0(\mathbf{x})}{f_1(\mathbf{x}) + f_0(\mathbf{x})} \right) \tag{17}$$

*is maximized only when $L(\mathbf{x}) = 0$ or $L(\mathbf{x}) = 1$ leading to $F(\mathbf{x}) = 0$. Therefore, the objective in Eq. (3) is maximized by setting $L(\mathbf{x}) = 0$ or $L(\mathbf{x}) = 1 \ \forall \mathbf{x} \in S_1 \cup S_2$.*

*Finally, since $L(\mathbf{x})$ is a continuous function, $\nexists \mathbf{x}_1, \mathbf{x}_2 \in S_1$ such that $L(\mathbf{x}_1) = 0$ and $L(\mathbf{x}_2) = 1$. We prove this by contradiction. Suppose there exists a pair $(\mathbf{x}_1, \mathbf{x}_2)$ of this kind. Then along any path connecting $\mathbf{x}_1$ and $\mathbf{x}_2$ within $S_1$, there must exist a point where $L(\mathbf{x})$ is not continuous since $L(\mathbf{x}) = 0$ or $L(\mathbf{x}) = 1 \ \forall \mathbf{x} \in S_1 \cup S_2$ to satisfy the maximization condition. This is a contradiction. By symmetry, the same argument can be proved for $\mathbf{x}_1, \mathbf{x}_2 \in S_2$. Therefore one of the two cases mentioned in the theorem must be the optimal solution for $L(\mathbf{x})$ in Eq. (3). Thus we have proved the claim.* $\square$

# B   NEURAL BAYES-DMS: EXTENSION TO MULTIPLE PARTITIONS

In order to extend our proposal to multiple partitions (say $K$), the idea is to find conditional distribution $q_i$ ($i \in [K]$) corresponding to each of the $K$ partitions such the divergence between conditional distribution of every partition and the conditional distribution of the combined remaining partitions is maximized. Specifically, we propose the following primary objective,

$$\max_{\substack{q_k \\ \pi_k \neq 0, \forall k \in [K]}} \frac{1}{K} \sum_{k=1}^{K} J(q_k(\mathbf{x}) \| \bar{q}_k(\mathbf{x})) \quad \text{s.t.} \tag{18}$$

$$\int_{\mathbf{x}} q_k(\mathbf{x}) = 1 \quad \forall k \in [K] \tag{19}$$

$$\sum_{k=1}^{K} q_k(\mathbf{x}) \cdot \pi_k = p(\mathbf{x}) \tag{20}$$

$$\sum_{k=1}^{K} \pi_k = 1 \tag{21}$$

where $\bar{q}_k(\mathbf{x})$ is the conditional distribution corresponding to the full data distribution excluding the partition defined by $q_k(\mathbf{x})$. Formally,

$$\bar{q}_k(\mathbf{x}) := \frac{p(\mathbf{x}) - q_k(\mathbf{x}) \cdot \pi_k}{1 - \pi_k} \tag{22}$$

Then the theorem below shows an equivalent way of solving the above objective.

**Theorem 2.** *Let $L(\mathbf{x}) : \mathbb{R}^n \to \mathbb{R}^{+^K}$ be a non-parametric function for any given input $\mathbf{x} \in \mathbb{R}^n$ with the property $\sum_{k=1}^{K} L_k(\mathbf{x}) = 1 \,\forall \mathbf{x}$, and let $J$ be the Jensen-Shannon divergence. Define scalars $f_k(\mathbf{x}) := \frac{L_k(\mathbf{x})}{\mathbb{E}_{\mathbf{x}}[L_k(\mathbf{x})]}$ and $\bar{f}_k(\mathbf{x}) := \frac{1 - L_k(\mathbf{x})}{1 - \mathbb{E}_{\mathbf{x}}[L_k(\mathbf{x})]}$. Then the objective in Eq. (18) is equivalent to,*

$$\max_{L_k \forall i \in [K]} \frac{1}{2} \cdot \mathbb{E}_{\mathbf{x}} \left[ \sum_{k=1}^{K} f_k(\mathbf{x}) \cdot \log \left( \frac{f_k(\mathbf{x})}{f_k(\mathbf{x}) + \bar{f}_k(\mathbf{x})} \right) + \bar{f}_k(\mathbf{x}) \cdot \log \left( \frac{\bar{f}_k(\mathbf{x})}{\bar{f}_k(\mathbf{x}) + f_k(\mathbf{x})} \right) \right] + \log 2 \tag{23}$$

*s.t.* $\quad \mathbb{E}_{\mathbf{x}}[L_k(\mathbf{x})] = \pi_k$

*Here $L_k(\mathbf{x})$ denotes the $k^{th}$ unit of $L(\mathbf{x})$.*

**Proof:** *Similar to theorem 1, the main idea is to parameterize $q_k$ and $\bar{q}_k$ as follows,*

$$q_k(\mathbf{x}) := \frac{L_k(\mathbf{x}) \cdot p(\mathbf{x})}{\mathbb{E}_{\mathbf{x} \sim p(\mathbf{x})}[L_k(\mathbf{x})]} \quad \bar{q}_k(\mathbf{x}) := \frac{(1 - L_k(\mathbf{x})) \cdot p(\mathbf{x})}{1 - \mathbb{E}_{\mathbf{x} \sim p(\mathbf{x})}[L_k(\mathbf{x})]} \tag{24}$$

*To verify that these parameterizations are valid, note that,*

$$\int_{\mathbf{x}} q_k(\mathbf{x}) = \int_{\mathbf{x}} \frac{L_k(\mathbf{x}) \cdot p(\mathbf{x})}{\mathbb{E}_{\mathbf{x} \sim p(\mathbf{x})}[L_k(\mathbf{x})]} = 1 \tag{25}$$

*Similarly, $\int_{\mathbf{x}} \bar{q}_k(\mathbf{x}) = 1$. To verify that the second constraint is satisfied, we use the above parameterization and substitute $\mathbb{E}_{\mathbf{x}}[L_k(\mathbf{x})] = \pi_k$ and get,*

$$\sum_{k=1}^{K} \frac{L_k(\mathbf{x}) \cdot p(\mathbf{x})}{\pi_k} \cdot \pi_k = p(\mathbf{x}) \cdot \left( \sum_{k=1}^{K} L_k(\mathbf{x}) \right) \tag{26}$$

$$= p(\mathbf{x}) \tag{27}$$

*where the last equality uses the definition of $L(\mathbf{x})$. Also notice that each $\pi_k \in [0, 1]$ and thus $\mathbb{E}_{\mathbf{x}}[L_k(\mathbf{x})] = \pi_k$ is feasible for any arbitrary distribution $q_k(\mathbf{x})$ when $L_k(\mathbf{x}) \geq 0$.*

*Finally, using the proposed parameterization we have,*

$$\bar{q}_k(\mathbf{x}) = \frac{p(\mathbf{x}) - q_i(\mathbf{x}) \cdot \pi_k}{1 - \pi_k} \tag{28}$$

$$= p(\mathbf{x}) \cdot \frac{1 - \frac{L^i(\mathbf{x})}{\mathbb{E}_\mathbf{x}[L_k(\mathbf{x})]} \cdot \pi_k}{1 - \pi_k} \tag{29}$$

$$= p(\mathbf{x}) \cdot \frac{1 - L^i(\mathbf{x})}{1 - \mathbb{E}_\mathbf{x}[L_k(\mathbf{x})]} \tag{30}$$

$$= \bar{f}_k(\mathbf{x}) \cdot p(\mathbf{x}) \tag{31}$$

*where we have used the fact that $\mathbb{E}_\mathbf{x}[L_k(\mathbf{x})] = \pi_k$. Using the definition of JS divergence, the max problem in Eq. (18) can be written as,*

$$\max_{L_k \forall i \in [K]} \frac{1}{2} \cdot \sum_{k=1}^{K} \int_\mathbf{x} q_k(\mathbf{x}) \cdot \log\left(\frac{q_k(\mathbf{x})}{0.5 \cdot (q_k(\mathbf{x}) + \bar{q}_k(\mathbf{x}))}\right) + \bar{q}_k(\mathbf{x}) \cdot \log\left(\frac{\bar{q}_k(\mathbf{x})}{0.5 \cdot (\bar{q}_k(\mathbf{x}) + q_k(\mathbf{x}))}\right) \tag{32}$$

*Substituting $q_k$ and $\bar{q}_k$ with their respective parameterizations and using the definitions of $f_k(\mathbf{x})$ and $\bar{f}_k(\mathbf{x})$ completes the proof.* □

In terms of implementation, we propose to simply have $K$ output units in the label generating network $L_\theta$ while sharing the rest of the network. Also, we use Softmax activation at the output layer to satisfy the properties of $L$ specified in the above theorem.

## C ADDITIONAL NEURAL BAYES-DMS EXPERIMENTS

We run an experiment on MNIST. We randomly split the training set into $90\% - 10\%$ training-validation set In this experiment, we train a CNN with the following architecture: $C(100, 3, 1, 0) - P(2, 2, 0, \max) - C(100, 3, 1, 0) - C(200, 3, 1, 0) - P(2, 2, 0, \max) - C(500, 3, 1, 0) - P(., ., ., \mathrm{avg}) - FC(10)$. Here $P(., ., ., \mathrm{avg})$ denotes the entire spatial field is average pooled to result in $1 \times 1$ height-width, and $FC(10)$ denotes a fully connected layer with output dimension 10. Finally, Softmax is applied at the output and the network is trained using the Neural Bayes-DMS objective. We optimize the objective using Adam with learning rate 0.001, batch size 5000, 0 weight decay for 100 epochs (other Adam hyper-parameters are kept standard). We use $\beta = 1$ for the smoothness regularization coefficient. Once this network is trained, we train a linear classifier on top of this 10 dimensional output using Adam with identical configurations except a batch size of 128 is used. We early stop on the validation set of MNIST and report the test accuracy using that model. The classifier reaches $99.22\%$ test accuracy. This experiment shows that MNIST classes lie on nearly disjoint manifolds and that Neural Bayes-DMS can correctly label them. As baseline, a linear classifier trained on features from a randomly initialized identical CNN architecture reaches $42.97\%$.

## D GRADIENT COMPUTATION PROBLEM FOR THE $\mathbb{E}_\mathbf{x}[L_\theta(\mathbf{x})]$ TERM

The Neural Bayes parameterization contains the term $\mathbb{E}_\mathbf{x}[L_\theta(\mathbf{x})]$. Computing unbiased gradient through this term is in general difficult without the use of very large batch-sizes even though the quantity $\mathbb{E}_\mathbf{x}[L_\theta(\mathbf{x})]$ itself may have a good estimate using very few samples. For instance, consider the scalar function $\psi(t) = 1 + 0.01 \sin \omega t$. Consider the scenario when $\omega \to \infty$. The quantity $\mathbb{E}[\psi(t)]$ can be estimated very accurately using even one example. Further, $\mathbb{E}[\psi(t)] = 1$, hence $\frac{\partial \mathbb{E}[\psi(t)]}{\partial t} = 0$. However, when using a finite number of samples, the approximation of $\frac{\partial \mathbb{E}[\psi(t)]}{\partial t}$ can have a very high variance estimate due to improper cancelling of gradient terms from individual samples.

In the case of Neural Bayes-MIM we found that gradients through terms involving $\mathbb{E}_\mathbf{x}[L_\theta(\mathbf{x})]$ were 0. This allows us to estimate gradients for this objective reliably in the mini-batch setting. But in general it may be challenging to do so and solving objectives using Neural Bayes parameterization may require a customized work-around for each objective.

## E    IMPLEMENTATION DETAILS OF THE NEURAL BAYES-MIM OBJECTIVE

We apply the Neural Bayes-MIM objective (Eq 10) to all the hidden layers at different scales (using average pooling). We now discuss its implementation details. Consider the CNN architecture used in our experiments– $C(200, 3, 1, 0) - P(2, 2, 0, \max) - C(500, 3, 1, 0) - C(700, 3, 1, 0) - P(2, 2, 0, \max) - C(1000, 3, 1, 0)$. Denote $\mathbf{h}^i$ ($i \in \{0, 1, 2, 3\}$) be the 4 hidden layer ReLU outputs after the 4 convolution layers. For input of size $32 \times 32 \times 3$, all these hidden states have height and width dimension in addition to channel dimension. For a mini-batch $B$, these hidden states are therefore 4 dimensional tensors. Let these 4 dimensions for the $i^{th}$ state be denoted by $|B| \times C_i \times H_i \times W_i$, where the dimensions denote batch-size, number of channels, height and width. Denote $\mathcal{S}$ to be the Softmax function applied along the channel dimension, and $\mathcal{P}$ to be $P(2, 2, 0, \text{avg})$. Further, denote $\mathbf{h}^i := \mathcal{P}(\mathbf{h}^{i-4})$ ($i \in \{4, 5, 6, 7\}$) as the scaled version of the original states computed by average pooling, and define numbers $C_i, H_i, W_i$ accordingly. Then the total Neural Bayes-MIM objective for this architecture is given by,

$$\min_\theta - \frac{1}{|B|} \sum_{\mathbf{x} \in B} \left[ \frac{1}{8} \sum_{i=0}^{7} \frac{1}{H_i W_i} \sum_{h,w=1}^{H_i, W_i} \sum_{k=1}^{C_i} \mathcal{S}(\mathbf{h}_{k,h,w}^i(\mathbf{x})) \log \langle \mathcal{S}(\mathbf{h}_{k,h,w}^i(\mathbf{x})) + \epsilon \rangle \right]$$
$$+ (1 + \alpha) \cdot \mathcal{R}_p(\theta) + \beta \cdot \mathcal{R}_c \tag{33}$$

where,

$$\mathcal{R}_p(\theta) := -\frac{1}{8} \sum_{i=0}^{7} \frac{1}{H_i W_i} \sum_{h,w=1}^{H_i, W_i} \left[ \sum_{k=1}^{C_i} \frac{1}{C_i} \log \left[ \frac{1}{|B|} \sum_{\mathbf{x} \in B} \mathcal{S}(\mathbf{h}_{k,h,w}^i(\mathbf{x})) \right] + \frac{C_i - 1}{C_i} \log \left[ 1 - \frac{1}{|B|} \sum_{\mathbf{x} \in B} \mathcal{S}(\mathbf{h}_{k,h,w}^i(\mathbf{x})) \right] \right] \tag{34}$$

and,

$$\mathcal{R}_c = \frac{1}{|B|} \sum_{\mathbf{x} \in B} \frac{\|\mathcal{P}(\mathbf{h}_k^3(\mathbf{x})) - \mathcal{P}(\mathbf{h}_k^3(\mathbf{x} + \zeta \cdot \hat{\delta}))\|^2}{\zeta^2} \tag{35}$$

where $\hat{\delta} := \frac{\delta}{\|\delta\|_2}$ is a normalized noise vector computed independently for each sample $\mathbf{x}$ in the batch $B$ as,

$$\delta := \mathbf{X}\mathbf{v}. \tag{36}$$

Here $\mathbf{X} \in \mathbb{R}^{n \times B}$ is the matrix containing the batch of samples, and each dimension of $\mathbf{v} \in \mathbb{R}^B$ is sampled i.i.d. from a standard Gaussian. This computation ensures that the perturbation lies in the span of data. Finally $\zeta$ is the scale of normalized noise added to all samples in a batch. In our experiments, since we always normalize the datasets to have zero mean and unit variance across all dimensions, we sample $\zeta \sim \mathcal{N}(0, 0.1^2)$. Note that for the architecture used, $\mathcal{P}(\mathbf{h}_k^3(\mathbf{x}))$ results in an output with height and width equal to 1, hence the output is effectively a 2D matrix of size $|B| \times C_3$. Finally, the gradient form this mini-batch is accumulated and averaged over multiple batches before updating the parameters for a more accurate estimate of gradients.

## F    ADDITIONAL ANALYSIS OF NEURAL BAYES-MIM

### F.1    GRADIENT STRENGTH OF UNIFORM PRIOR IN NEURAL BAYES-MIM-V1 (EQ 8) VS NEURAL BAYES-MIM-V2 (9)

As discussed in the main text, the term,

$$\mathcal{R}_p^{v1}(\theta) := \sum_{k=1}^{K} \mathbb{E}_{\mathbf{x}}[L_k(\mathbf{x})] \log \langle \mathbb{E}_{\mathbf{x}}[L_k(\mathbf{x})] \rangle \tag{37}$$

acts as a uniform prior encouraging the representations to be distributed. However, gradients are much stronger when $\mathbb{E}_{\mathbf{x}}[L_k(\mathbf{x})]$ approaches 1 for the alternative cross-entropy formulation,

$$\mathcal{R}_p^{v2}(\theta) := -\sum_{k=1}^{K} \frac{1}{K} \log \mathbb{E}_{\mathbf{x}}[L_k(\mathbf{x})] + \frac{K-1}{K} \log(1 - \mathbb{E}_{\mathbf{x}}[L_k(\mathbf{x})]) \tag{38}$$

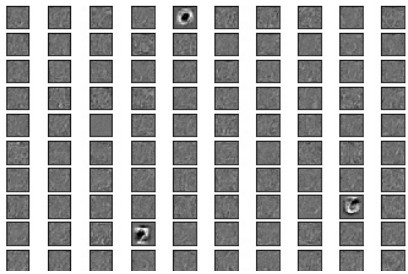

Figure 3: MNIST filters learned using Neural Bayes-MIM-v1 objective (Eq 8) using the configuration $\alpha = 4$, $\beta = 4$. Majority of filters are dead. For comparison with filters from Neural Bayes-MIM-v2, see figure 4 (bottom).

To see this, note that gradient for $\mathcal{R}_p^{v1}(\theta)$ is given by,

$$\frac{\partial \mathcal{R}_p^{v1}(\theta)}{\partial \theta} = \sum_{k=1}^{K} \frac{\partial \mathbb{E}_{\mathbf{x}}[L_{\theta_k}(\mathbf{x})]}{\partial \theta} \log \mathbb{E}_{\mathbf{x}}[L_{\theta_k}(\mathbf{x})] - \sum_{k=1}^{K} \frac{\partial \mathbb{E}_{\mathbf{x}}[L_{\theta_k}(\mathbf{x})]}{\partial \theta} \tag{39}$$

$$= \sum_{k=1}^{K} \frac{\partial \mathbb{E}_{\mathbf{x}}[L_{\theta_k}(\mathbf{x})]}{\partial \theta} \log \mathbb{E}_{\mathbf{x}}[L_{\theta_k}(\mathbf{x})] - \frac{\partial \mathbb{E}_{\mathbf{x}}[\sum_{k=1}^{K} L_{\theta_k}(\mathbf{x})]}{\partial \theta} \tag{40}$$

$$= \sum_{k=1}^{K} \frac{\partial \mathbb{E}_{\mathbf{x}}[L_{\theta_k}(\mathbf{x})]}{\partial \theta} \log \mathbb{E}_{\mathbf{x}}[L_{\theta_k}(\mathbf{x})] \tag{41}$$

where the last equality holds due to the linearity of expectation and because $\sum_{k=1}^{K} L_{\theta_k}(\mathbf{x}) = 1$ by design. On the other hand, gradients for $\mathcal{R}_p^{v2}(\theta)$ is given by,

$$\frac{\partial \mathcal{R}_p^{v2}(\theta)}{\partial \theta} = -\sum_{k=1}^{K} \frac{1}{K} \left( \frac{1}{\mathbb{E}_{\mathbf{x}}[L_k(\mathbf{x})]} - \frac{K-1}{1 - \mathbb{E}_{\mathbf{x}}[L_k(\mathbf{x})]} \right) \frac{\partial \mathbb{E}_{\mathbf{x}}[L_{\theta_k}(\mathbf{x})]}{\partial \theta} \tag{42}$$

When the representation being learned is such that the marginal $p(z)$ peaks along a single state $k$, i.e., $\mathbb{E}_{\mathbf{x}}[L_k(\mathbf{x})] \to 1$ (making the representation degenerate), the gradient for the $k^{th}$ term for v1 is given by,

$$\frac{\partial \mathbb{E}_{\mathbf{x}}[L_{\theta_k}(\mathbf{x})]}{\partial \theta} \log \mathbb{E}_{\mathbf{x}}[L_{\theta_k}(\mathbf{x})] \approx 0 \tag{43}$$

while that for v2 is given by,

$$-\frac{1}{K} \left( \frac{1}{\mathbb{E}_{\mathbf{x}}[L_k(\mathbf{x})]} - \frac{K-1}{1 - \mathbb{E}_{\mathbf{x}}[L_k(\mathbf{x})]} \right) \frac{\partial \mathbb{E}_{\mathbf{x}}[L_{\theta_k}(\mathbf{x})]}{\partial \theta} \approx \lim_{c \to 0} \frac{1}{c} \cdot \frac{\partial \mathbb{E}_{\mathbf{x}}[L_{\theta_k}(\mathbf{x})]}{\partial \theta} \tag{44}$$

whose magnitude approaches infinity as $\mathbb{E}_{\mathbf{x}}[L_k(\mathbf{x})] \to 1$. Thus $\mathcal{R}_p^{v2}(\theta)$ is beneficial in terms of gradient strength.

## F.2 EMPIRICAL COMPARISON BETWEEN NEURAL BAYES-MIM-v1 (EQ 8) AND NEURAL BAYES-MIM-v2 (9)

To empirically understand the difference in behavior of Neural Bayes-MIM objective v1 vs v2, we first plot the filters learned by the v1 objective and compare it with those learned by the v2 objective. The filters learned by the v1 objective are shown in figure 3 using the configuration $\alpha = 4$, $\beta = 4$. It can be seen that most filters are dead. We tried other configurations as well without any change in the outcome. Since the v1 and v2 objective differ only in the formulation of the uniform prior regularization, as explained in the previous section, we believe that v1 leads to dead filters because of weak gradients from its regularization term.

In the second set of experiments, we train many models using Neural Bayes-MIM-v1 and Neural Bayes-MIM-v2 objectives separately with different hyper-parameter configurations similar to the setting of figure 2. The performance scatter plot is shown in figure 6. We find that Neural Bayes-MIM-v2 has better average and best performance compared with Neural Bayes-MIM-v1.

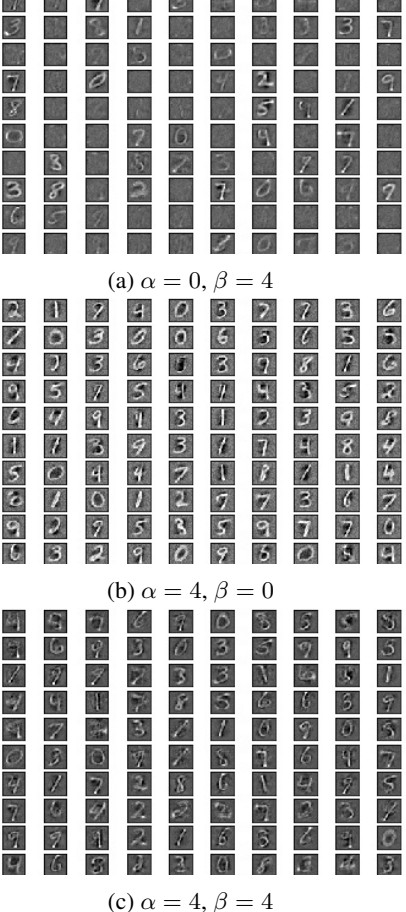

(a) $\alpha = 0$, $\beta = 4$

(b) $\alpha = 4$, $\beta = 0$

(c) $\alpha = 4$, $\beta = 4$

Figure 4: MNIST filters learned using Neural Bayes-MIM objective Eq 10. Majority of filters are dead when regularization coefficient $\alpha = 0$. Filters memorize input samples when regularization coefficient $\beta = 0$. Using both regularization terms results in filters that mainly capture parts of inputs which are good for distributed representation.

| MBS \BS | 50 | 250 | 500 | 2000 | 3000 |
|---------|-------|-------|-------|-------|-------|
| 50 | 40.62 | 42.97 | 41.41 | 75 | 78.91 |
| 100 | N/A | 67.97 | 66.41 | 78.12 | 78.91 |
| 250 | N/A | 76.56 | 78.91 | 82.03 | 84.38 |
| 500 | N/A | N/A | 82.03 | 78.91 | 79.69 |

Table 2: Effect of mini-batch size (MBS) and batch size (BS) on final test accuracy of Neural Bayes-MIM on CIFAR-10. Gradients are computed using batches of size MBS and accumulated until BS samples are seen before parameter update. As expected, a sufficiently large MBS is needed for computing high fidelity gradients due to $\mathbb{E}_{\mathbf{x}}[L_k(\mathbf{x})]$ term. Gradient accumulation using BS further helps. All models are trained for the same number of *epochs*.

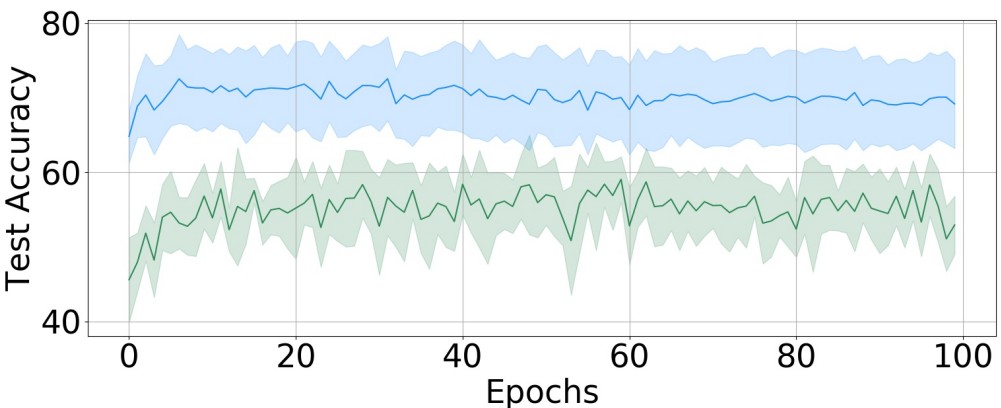

Figure 5: Mean (solid line) and standard deviation (error band) of accuracy evolution during MLP classifier training on CIFAR-10 using Neural Bayes-MIM encoder (blue) and random encoder (green). Features learned by Neural Bayes-MIM allow training to start at a much higher value ($\sim 65\%$ on average) and converge faster as opposed to a random encoder ($\sim 47\%$ on average).

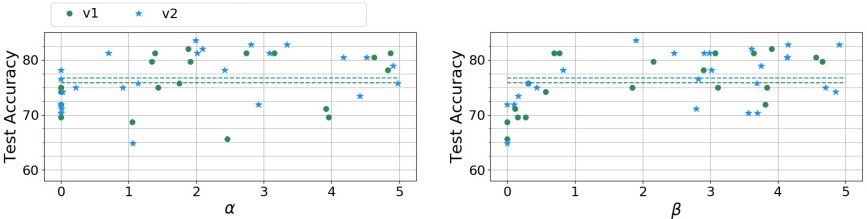

Figure 6: Performance of Neural Bayes-MIM-v1 vs Neural Bayes-MIM-v2. A CNN encoder is trained using Neural Bayes-MIM with different configurations– hyper-parameters $\alpha$, $\beta$ and scaling of states on which the objective is applied. A one hidden layer classifier (with 200 units) is then trained using labels on these frozen features to get the final test accuracy.Green and blue dotted lines are the average of all the green and blue points respectively. Neural Bayes-MIM-v2 has better average and best performance compared with Neural Bayes-MIM-v1.

## F.3 Additional Experiments

**Visualization of Filters**: We visualize the filters learned by the Neural Bayes-MIM objective on MNIST digits and qualitatively study the effects of the regularizations used. For this we train a deep fully connected network with 3 hidden layers each of width 500 using Adam with learning rate 0.001, batch size 500, 0 weight decay for 50 epochs (other Adam hyper-parameters are kept standard). We train three configurations: 1. $\alpha = 0$, $\beta = 4$; 2. $\alpha = 4$, $\beta = 0$; 3. $\alpha = 4$, $\beta = 4$. The learned filters are shown in figure 3. We find that the uniform prior regularization ($\alpha > 0$) prevents dead filters while the smoothness regularization ($\beta > 0$) prevents input memorization.

**Accuracy vs Epochs**: Finally, we plot the evolution of accuracy over epochs for all the models learned in the experiments of figure 2. For Neural Bayes-MIM we use the models with scaling (42 in total), and all 16 models for the random encoder. The convergence plot (figure 5) shows that models pre-trained with Neural Bayes-MIM quickly converge to a higher test accuracy compared to the baseline.

**Effect of Mini-batch size (MBS) and Batch size (BS)**: During implementation, we proposed to compute gradients using a reasonably large mini-batch of size MBS and accumulate gradients until BS samples are seen. This is done to overcome the gradient estimation problem due to the $\mathbb{E}_{\mathbf{x}}[L_k(\mathbf{x})]$ term in Neural Bayes-MIM. Here we evaluate the effect of these two hyper-parameters on the final test performance. We choose MBS from $\{50, 100, 250, 500\}$ and BS from $\{50, 250, 500, 2000, 3000\}$.

For each combination of MBS and BS, we train the CNN encoder using Neural Bayes-MIM with $\alpha = 2$ and $\beta = 4$ (chosen by examining figure 2); the rest of the training settings are kept identical to those used for figure 2 experiment. Table 2 shows the final test accuracy on CIFAR-10 for each combination of hyper-parameters MBS and BS. We make two observations: 1. using very small MBS (Eg. 50 and 100) typically results in poor (even worse than that of a random encoder (67.97%)), while larger MBS significantly improves performance; 2. using a larger BS further improves performance in most cases (even when MBS is small).

# G  PROOF OF LEMMA 1

**Lemma 1.** *Let $p(\mathbf{x}|z = k)$ and $p(z)$ be any conditional and marginal distribution defined for continuous random variable $\mathbf{x}$ and discrete random variable $z$. If $\mathbb{E}_{\mathbf{x} \sim p(\mathbf{x})}[L_k(\mathbf{x})] \neq 0 \; \forall k \in [K]$, then there exists a non-parametric function $L(\mathbf{x}) : \mathbb{R}^n \to \mathbb{R}^{+^K}$ for any given input $\mathbf{x} \in \mathbb{R}^n$ with the property $\sum_{k=1}^{K} L_k(\mathbf{x}) = 1 \; \forall \mathbf{x}$ such that,*

$$p(\mathbf{x}|z = k) = \frac{L_k(\mathbf{x}) \cdot p(\mathbf{x})}{\mathbb{E}_{\mathbf{x} \sim p(\mathbf{x})}[L_k(\mathbf{x})]}, \quad p(z = k) = \mathbb{E}_{\mathbf{x}}[L_k(\mathbf{x})], \quad p(z = k|\mathbf{x}) = L_k(\mathbf{x}) \tag{45}$$

*and this parameterization is consistent.*

**Proof**: *First we show the existence proof. Notice that there exists a non-parametric function $g_k(\mathbf{x}) := \frac{p(\mathbf{x}|z=k)}{p(\mathbf{x})} \; \forall \mathbf{x} \in supp(p(\mathbf{x}))$. Denote $G_k(\mathbf{x}) = p(z = k)g_k(\mathbf{x})$. Then,*

$$\mathbb{E}_{\mathbf{x}}[G_k(\mathbf{x})] = \mathbb{E}_{\mathbf{x}}[p(z = k)g_k(\mathbf{x})] = p(z = k) \tag{46}$$

*and,*

$$\frac{G_k(\mathbf{x})}{\mathbb{E}_{\mathbf{x}}[G_k(\mathbf{x})]} = \frac{p(z = k)g_k(\mathbf{x})}{p(z = k)} = \frac{p(\mathbf{x}|z = k)}{p(\mathbf{x})} \tag{47}$$

*Thus $L_k := G_k$ works. To verify that this parameterization is consistent, note that for any $k$,*

$$\int_{\mathbf{x}} p(\mathbf{x}|z = k) = \int_{\mathbf{x}} \frac{L_k(\mathbf{x}) \cdot p(\mathbf{x})}{\mathbb{E}_{\mathbf{x} \sim p(\mathbf{x})}[L_k(\mathbf{x})]} = 1 \tag{48}$$

*where we use the condition $\mathbb{E}_{\mathbf{x} \sim p(\mathbf{x})}[L_k(\mathbf{x})] \neq 0 \; \forall k \in [K]$. Secondly, we note that,*

$$\sum_{k=1}^{K} p(\mathbf{x}|z = k) \cdot p(z = k) = \sum_{k=1}^{K} \frac{L_k(\mathbf{x}) \cdot p(\mathbf{x})}{\mathbb{E}_{\mathbf{x}}[L_k(\mathbf{x})]} \cdot \mathbb{E}_{\mathbf{x}}[L_k(\mathbf{x})] \tag{49}$$

$$= \sum_{k=1}^{K} L_k(\mathbf{x}) \cdot p(\mathbf{x}) \tag{50}$$

$$= p(\mathbf{x}) \tag{51}$$

*where the last equality is due to the conditions $\sum_{k=1}^{K} L_k(\mathbf{x}) = 1 \; \forall \mathbf{x}$. Thirdly,*

$$\sum_{k=1}^{K} p(z = k) = \sum_{k=1}^{K} \mathbb{E}_{\mathbf{x}}[L_k(\mathbf{x})] \tag{52}$$

$$= \mathbb{E}_{\mathbf{x}}\left[\sum_{k=1}^{K} L_k(\mathbf{x})\right] \tag{53}$$

$$= 1$$

*Finally, we have from Bayes' rule:*

$$p(z = k|\mathbf{x}) = \frac{p(\mathbf{x}|z = k) \cdot p(z = k)}{p(\mathbf{x})} \tag{54}$$

$$= \frac{L_k(\mathbf{x}) \cdot p(\mathbf{x})}{\mathbb{E}_{\mathbf{x} \sim p(\mathbf{x})}[L_k(\mathbf{x})]} \cdot \frac{\mathbb{E}_{\mathbf{x} \sim p(\mathbf{x})}[L_k(\mathbf{x})]}{p(\mathbf{x})} \tag{55}$$

$$= L_k(\mathbf{x}) \tag{56}$$

*where the second equality holds because of the existence and consistency proofs of $p(\mathbf{x}|z = k) := \frac{L_k(\mathbf{x}) \cdot p(\mathbf{x})}{\mathbb{E}_{\mathbf{x} \sim p(\mathbf{x})}[L_k(\mathbf{x})]}$ and $p(z = k) := \mathbb{E}_{\mathbf{x}}[L_k(\mathbf{x})]$ shown above.* $\square$

# H    PROOFS FOR NEURAL BAYES-MIM

**Proposition 2.** *(Neural Bayes-MIM-v1) (proposition 1 in main text) Let $L(\mathbf{x}) : \mathbb{R}^n \to \mathbb{R}^{+K}$ be a non-parametric function for any given input $\mathbf{x} \in \mathbb{R}^n$ with the property $\sum_{i=1}^K L_k(\mathbf{x}) = 1 \ \forall \mathbf{x}$. Consider the following objective,*

$$L^* = \arg\max_L \mathbb{E}_{\mathbf{x}} \left[ \sum_{k=1}^K L_k(\mathbf{x}) \log \frac{L_k(\mathbf{x})}{\mathbb{E}_{\mathbf{x}}[L_k(\mathbf{x})]} \right] \tag{57}$$

*Then $L_k^*(\mathbf{x}) = p(z^* = k|\mathbf{x})$, where $z^* \in \arg\max_z MI(\mathbf{x}, z)$.*

**Proof**: *Using the Neural Bayes parameterization in lemma 1, we have,*

$$MI(\mathbf{x}, z) = \int_{\mathbf{x}} \sum_{k=1}^K p(\mathbf{x}, z = k) \log \frac{p(\mathbf{x}, z = k)}{p(\mathbf{x})p(z = k)} \tag{58}$$

$$= \int_{\mathbf{x}} \sum_{k=1}^K p(z = k|\mathbf{x})p(\mathbf{x}) \log \frac{p(z = k|\mathbf{x})}{p(z = k)} \tag{59}$$

$$= \int_{\mathbf{x}} \sum_{k=1}^K L_k(\mathbf{x}) \cdot p(\mathbf{x}) \cdot \log \frac{L_k(\mathbf{x})}{\mathbb{E}_{\mathbf{x}}[L_k(\mathbf{x})]} \tag{60}$$

$$= \mathbb{E}_{\mathbf{x} \sim p(\mathbf{x})} \left[ \sum_{k=1}^K L_k(\mathbf{x}) \log \frac{L_k(\mathbf{x})}{\mathbb{E}_{\mathbf{x}}[L_k(\mathbf{x})]} \right] \tag{61}$$

*Therefore the two objectives are equivalent and we have a closed form estimate of mutual information. Given $z^*$ is a maximizer of $MI(\mathbf{x}, z)$, since $L$ is a non-parametric function, there exists $L^*$ such that $p(z^* = k|\mathbf{x}) = L_k^*(\mathbf{x})$ due to lemma 1.* $\square$

**Theorem 3.** *(Theorem 1 in main text) Denote,*

$$J(\theta) = -\mathbb{E}_{\mathbf{x}} \left[ \sum_{k=1}^K L_{\theta_k}(\mathbf{x}) \log \frac{L_{\theta_k}(\mathbf{x})}{\mathbb{E}_{\mathbf{x}}[L_{\theta_k}(\mathbf{x})]} \right] \tag{62}$$

$$\hat{J}(\theta) = -\mathbb{E}_{\mathbf{x}} \left[ \sum_{k=1}^K L_{\theta_k}(\mathbf{x}) \log \langle \frac{L_{\theta_k}(\mathbf{x})}{\mathbb{E}_{\mathbf{x}}[L_{\theta_k}(\mathbf{x})]} \rangle \right] \tag{63}$$

*where $\langle . \rangle$ denotes gradients are not computed through the argument. Then $\frac{\partial J(\theta)}{\partial \theta} = \frac{\partial \hat{J}(\theta)}{\partial \theta}$.*

**Proof**: *We note that,*

$$J(\theta) = -\mathbb{E}_{\mathbf{x}} \left[ \sum_{k=1}^K L_{\theta_k}(\mathbf{x}) \log \frac{L_{\theta_k}(\mathbf{x})}{\mathbb{E}_{\mathbf{x}}[L_{\theta_k}(\mathbf{x})]} \right] \tag{64}$$

$$= -\mathbb{E}_{\mathbf{x}} \left[ \sum_{k=1}^K L_{\theta_k}(\mathbf{x}) \log L_{\theta_k}(\mathbf{x}) \right] + \mathbb{E}_{\mathbf{x}} \left[ \sum_{k=1}^K L_{\theta_k}(\mathbf{x}) \log \mathbb{E}_{\mathbf{x}}[L_{\theta_k}(\mathbf{x})] \right] \tag{65}$$

*Denote the first term by $T_1$. Then due to chain rule,*

$$-\frac{\partial T_1}{\partial \theta} = \mathbb{E}_{\mathbf{x}} \left[ \sum_{k=1}^K \frac{\partial L_{\theta_k}(\mathbf{x})}{\partial \theta} \log L_{\theta_k}(\mathbf{x}) \right] - \mathbb{E}_{\mathbf{x}} \left[ \sum_{k=1}^K \frac{L_{\theta_k}(\mathbf{x})}{L_{\theta_k}(\mathbf{x})} \cdot \frac{\partial L_{\theta_k}(\mathbf{x})}{\partial \theta} \right] \tag{66}$$

$$= \mathbb{E}_{\mathbf{x}} \left[ \sum_{k=1}^K \frac{\partial L_{\theta_k}(\mathbf{x})}{\partial \theta} \log L_{\theta_k}(\mathbf{x}) \right] - \mathbb{E}_{\mathbf{x}} \left[ \sum_{k=1}^K \frac{\partial L_{\theta_k}(\mathbf{x})}{\partial \theta} \right] \tag{67}$$

$$= \mathbb{E}_{\mathbf{x}} \left[ \sum_{k=1}^K \frac{\partial L_{\theta_k}(\mathbf{x})}{\partial \theta} \log L_{\theta_k}(\mathbf{x}) \right] - \mathbb{E}_{\mathbf{x}} \left[ \frac{\partial \sum_{k=1}^K L_{\theta_k}(\mathbf{x})}{\partial \theta} \right] \tag{68}$$

$$= \mathbb{E}_{\mathbf{x}} \left[ \sum_{k=1}^K \frac{\partial L_{\theta_k}(\mathbf{x})}{\partial \theta} \log L_{\theta_k}(\mathbf{x}) \right] \tag{69}$$

*where the last equality holds due to the linearity of expectation and because $\sum_{k=1}^{K} L_{\theta_k}(\mathbf{x}) = 1$ by design. Now denote the second term by $T_2$. Then due to chain rule,*

$$-\frac{\partial T_2}{\partial \theta} = -\mathbb{E}_{\mathbf{x}} \left[ \sum_{k=1}^{K} \frac{\partial L_{\theta_k}(\mathbf{x})}{\partial \theta} \log \mathbb{E}_{\mathbf{x}}[L_{\theta_k}(\mathbf{x})] \right] - \mathbb{E}_{\mathbf{x}} \left[ \sum_{k=1}^{K} \frac{L_{\theta_k}(\mathbf{x})}{\mathbb{E}_{\mathbf{x}}[L_{\theta_k}(\mathbf{x})]} \cdot \frac{\partial \mathbb{E}_{\mathbf{x}}[L_{\theta_k}(\mathbf{x})]}{\partial \theta} \right] \quad (70)$$

$$= -\mathbb{E}_{\mathbf{x}} \left[ \sum_{k=1}^{K} \frac{\partial L_{\theta_k}(\mathbf{x})}{\partial \theta} \log \mathbb{E}_{\mathbf{x}}[L_{\theta_k}(\mathbf{x})] \right] - \sum_{k=1}^{K} \frac{\mathbb{E}_{\mathbf{x}}[L_{\theta_k}(\mathbf{x})]}{\mathbb{E}_{\mathbf{x}}[L_{\theta_k}(\mathbf{x})]} \cdot \frac{\partial \mathbb{E}_{\mathbf{x}}[L_{\theta_k}(\mathbf{x})]}{\partial \theta} \quad (71)$$

$$= -\mathbb{E}_{\mathbf{x}} \left[ \sum_{k=1}^{K} \frac{\partial L_{\theta_k}(\mathbf{x})}{\partial \theta} \log \mathbb{E}_{\mathbf{x}}[L_{\theta_k}(\mathbf{x})] \right] - \sum_{k=1}^{K} \mathbb{E}_{\mathbf{x}} \left[ \frac{\partial L_{\theta_k}(\mathbf{x})}{\partial \theta} \right] \quad (72)$$

$$= -\mathbb{E}_{\mathbf{x}} \left[ \sum_{k=1}^{K} \frac{\partial L_{\theta_k}(\mathbf{x})}{\partial \theta} \log \mathbb{E}_{\mathbf{x}}[L_{\theta_k}(\mathbf{x})] \right] - \mathbb{E}_{\mathbf{x}} \left[ \frac{\partial \sum_{k=1}^{K} L_{\theta_k}(\mathbf{x})}{\partial \theta} \right] \quad (73)$$

$$= -\mathbb{E}_{\mathbf{x}} \left[ \sum_{k=1}^{K} \frac{\partial L_{\theta_k}(\mathbf{x})}{\partial \theta} \log \mathbb{E}_{\mathbf{x}}[L_{\theta_k}(\mathbf{x})] \right] \quad (74)$$

*where once again the last equality holds due to the linearity of expectation and because $\sum_{k=1}^{K} L_{\theta_k}(\mathbf{x}) = 1$ by design. Thus the gradient for $J$ is given by,*

$$\frac{\partial J(\theta)}{\partial \theta} = -\mathbb{E}_{\mathbf{x}} \left[ \sum_{k=1}^{K} \frac{\partial L_{\theta_k}(\mathbf{x})}{\partial \theta} \log \frac{L_{\theta_k}(\mathbf{x})}{\mathbb{E}_{\mathbf{x}}[L_{\theta_k}(\mathbf{x})]} \right] \quad (75)$$

*which is the same as $\frac{\partial \hat{J}(\theta)}{\partial \theta}$. This concludes the proof.* $\square$

