# OpenReview forum: "Neural Bayes: A Generic Parameterization Method for Unsupervised Learning"
_ICLR.cc/2021/Conference — Reject_

### Official Review · AnonReviewer4 · 2020-10-27
**Potentially useful clustering loss functions, but the paper is poorly written in both description and technical details**

**Rating:** 4
**Confidence:** 3

**Review:**

# Summary
The paper introduces a new function $L(x)$ so that, when optimised under certain objectives defined over continuous observation $x$ and discrete latent $z$, learns the correct clustering probability $p(z|x)$. The loss functions considered are the Jensen-Fisher divergence and muture information. The authors introduces modifications to the principled objectives in practice and demonstrate performance on toy and real image datasets.

# Pros:
* The functional view on learning the clustering function $p(z|x)$ is interesting
* This paper explores a few theoretically motivated new unsupervised objectives and practical regularization, all of which seem new and are worth further investigation
* Thm1 and Thm2 are correct and interesting.

# Cons:
* The paper is presented in a mysterious language both in the qualititative description and mathematical details. The reader would have to do a lot of guesswork and think a lot to understand what the paper is trying to say (not yet to verify the theoretical statements). I will ask the authors to clarify a lot of them later, and I'm happy to raise the score on satisfactory feedback
* The heuristics introduced for practical use is not accompanied with enough (though some) justifications other than "we tried a few"
* The experiments seem to require a lot of hand-tuning of hyperparameters and eyeballing for good results, making the methods less useful in practice.

# Recommendation:
I think this paper is not ready for publication, both in terms of clarity and impact. The idea is worth investigating more.

# Issues and questions:
1. The notation and statement in the fundamental Lemma 1 is unclear. Are $p(x|z)$ and $p(z)$ from the same joint distribution $p(x,z)$? If so, then they would imply a consistent distribution $p(x)$ which is the data distribution, and the results hold trivially. If they are not from a consistent joint, then the construction of $G_k$ in the proof at Appendix G does not guarantee a normalised distribution such that $\sum_k G_k(x) = 1$, and the equality of (51) does not hold. Can the authors clarify this, ideally using different notations for each distribution, e.g. $p(z)$, $q(x|z)$ and $pi(x)$ to avoid confusion.
1. If "if... then...." statement is unclear. How can $L$ be involved in the "if" part, and also "exist" as a consequence in the "then" part?
1. Is the normalisation property $\sum_k L_k(x) = 1$ a condition or a consequence?
1. In the definition of connected set, what is a path?
1. In the proof of Prop 1, why are the constructions of $q_1$ and $q_2$ optimal for the optimisation problem (2)? I think it may be plausible, but from a reader's perspective, I hope to see a clear proof of this.
1. Can the author provide more justification for the smoothness regularisation (4)? In particular, why is the perturbation in the span of data?
1. In the experiments at 4.2, I think we would really like to see the effect of regularisation weights on the results, andThe line "Regularization coefficient β was chosen from [0.5, 6] that resulted in optimal clustering" seems to suggest that eyeballing on the results is needed, which is only possible for 1- or 2-D problems.
1. Is there a way to choose the number of clusters?
1. The authors claims that DMS finds manifolds of subspaces, but in fact it learns a clustering function which does not identify the manifold structure.
1. Prop 2 raises multiple confusions if not inaccuracies. The objective cannot be written as an argmax. Mutual information is between two random variables rather than symbols (What is $z^*$?).
1.  In the proof of Prop 2, I do not follow the last sentence. I understand (58)-(61), but why is the posterior the optimal solution to (57)?
1. What's the connection between the two formualitons? Is it correct that MIM does not require the smoothness assumption needed for DMS?

# Detailed comments:
1. In the paragraph above Section 4, "through which computing the gradient is infeasbile", is this the gradient w.r.t. $\theta$? I also do not follow the example you listed in the appendix at all. Does $t$ play the same role as $x$?
1. 7th line in the 2nd to last paragraph of Section 2, "both both"
1. Please use bold for all vectors, such as $\delta$ in (4).
1. In the MNIST experiment of 4.2, how do other clustering algorithms perform in downstream linear classification?
1. Why is (9) called a cross-entropy formulation? Is the minus in (9) sign a typo? Also, $L_k$ should be $L_{\theta,k}$ here and also in other places.
1. The comment following (9) is not very obvious at least to me. The reader would have to check it themselves. (There is an extra comma after "both")
1. The comment "This is an attribute of distributed representation which is a fundamental goal in deep learning." is questionable: why is a evenly distributed representation a fundamental goal of deep learning?


===== update =====
No response is provided so I am maintaining the score. I hope the authors could address all the issues in a future version.

---

### Official Review · AnonReviewer3 · 2020-10-27
**Neat Ideas, but Experiments need more work**

**Rating:** 4
**Confidence:** 4

**Review:**


Summary.
The paper proposes Neural Bayes, a special parameterization of the posterior p(z|x) in Bayes' theorem with a neural network when the latent variables z are discrete. It presents two further developments of this idea, that each rely on different assumption on the latent code: 1) Disjoint Manifold Separation (DMS) assumes disjoint support of the conditional distributions p(x|z), and can is shown to be usable for clustering; 2) Mutual Information Maximization (MIM) maximizes the mutual information of x and z, and is shown to (unsupervisedly) generate feature representations, which when combined with an supervised MLP outperform another method on CIFAR-10, CIFAR-100 & STL-10.

Reasons for score:
In general the applicability of the proposed methods remains unclear - or where it is given (e.g. clustering, or unsupervised feature extraction), insufficient experimental evidence for its usefulness is given. Hence, I vote for rejection of the paper in its current form.

Pros:
- The disjoint manifold separation method is a novel idea to me, and is a very reasonable approach to cluster data
- MIM is a promising approach to learn unsupervised feature representations
- the Gradient Simplification is a nice trick to reduce variance of estimates for the MIM


Cons:
- Experiments of DMS needs more elaboration:
	- how does this behave when the number of possible z is not equal the number of clusters in the data?
	- how does the method compare to other clustering methods (e.g. spectral clustering, DBSCAN, or other state-of-the art methods?)
- Experiments of MIM:
   - comparison to other self-supervised methods is necessary (e.g. SimCLR or SimCLRv2) to put the method in context to the state-of-the art here, as the only shown application is fitting this context.
   - the best performing model is chosen; this is not good practice, as it can be seen in Figure 2 that the variance of the performance is very high, hence running more models will also yield better performing models. Please report mean + std-dev (or some other reasonable statistic) for a given parameter set (alpha, beta,..) over a reasonable size of runs.
- Formal exposition could be clearer, e.g. instead of L_k one could directly call this p(z|k) and mention that it's parametrized by a NN (p_theta(.) then). Or in chapter 4, why is q_i(x) introduced as a "conditional distribution" - one could just use p(x|z).

Rebuttal:
- please address my points in Cons.

Other small comments:
- in the abstract: "...express p(xjz), p(zjx) and p(z) in closed form". As written later in the text, Monte-Carlo is needed to calculate p(z) etc. (it's an integral over x), so exact evaluation is not possible in a finite number of operations, and hence it's not closed form.
- the smoothness of L_theta is reminiscent of recent developments showing that Lipschitz continuity regularization is beneficial for generalization. The relation should be worked out.

---

### Official Review · AnonReviewer2 · 2020-10-28
**Some interesting results but the empirical results are limited**

**Rating:** 5
**Confidence:** 4

**Review:**

This work introduces a parameterization called Neural Bayes that facilitates learning representations from unlabeled data by categorizing them, where each data point x is mapped to a latent discrete variable z such that the distribution p(x) is segmented into a finite number of conditional distributions. Imposing different constraints on the latent discrete space will result in learning representations manifesting various properties. Two use cases of the proposed parameterization are studied: disjoint manifold separation and mutual information maximization.

As argued by the authors, the disjoint manifold separation objective and the optimality as shown in Theorem 1 is very interesting. Such a simple objective can lead to optimal manifold separation but this may also due to the strong restriction of the disjoint manifold assumption. Unfortunately, the authors do not elaborate much on this and only present some preliminary results on some synthetic datasets without much discussion. The results in Figure 1 do not show much benefits of the proposed method without a proper comparison to existing deep subspace clustering methods. Also, it is argued in Section 1 that "imposing different conditions on the latent space z through different objective functions will result in learning qualitatively different representations" which is the central theme of this work, it would be nice to cast the DMS objective into this framework as well to show how equation (3) corresponds to a regularization on z.

On the other hand, the proposed Neural Bayes framework looks identical to a standard mixture model and the analysis follows naturally from the Bayes' rule. Similar ideas of unsupervised representation learning via mixture models have been explored extensively in the literature, DeepCluster, IMSAT, VQ-VAE, to name a few. Almost all methods share the same mixture framework, but use different regularizations on the conditionals or the latent discrete variables. The Neural Bayes-MIM-v1 objective is identical to the IMSAT method as discussed by the authors. This work presents a better theoretical justification, but actual objective used (10) is different from theoretical-backed objective (8) minus the smooth regularization. When $\alpha \ne 0$, (10) is different from the optimal result. It would be nice to discuss the tradeoff manifested by controlling the entropy and how it impacts the learned representation and the downstream tasks. It also would be good to include the quantitative comparison of (8) and (9) in the main text to give a complete story.  As in the disjoint manifold separation case, the quantitative results are weak and only selected baselines are compared. In the description of the settings, only the architecture and learning parameters are discussed, but the number of clusters used is never mentioned. What K is used in the experimental settings? What happens if we overcluster the data which is usually the case in practice since we have no idea about the true number of clusters. This might create a serious problem for the disjoint manifold separation case due to the strong disjoint assumptions.

To conclude, this work seems preliminary and I do not think it passes the bar for acceptance in its current format.

---

### Official Review · AnonReviewer1 · 2020-10-29
**Novelty is well below the ICLR level**

**Rating:** 5
**Confidence:** 5

**Review:**

- Novelty is minimal, given previous work in clustering, latent variable models, etc.

- The structure of the paper (mainly with the two use cases at the core) is rather non-standard. Regardless of that, the ideas proposed in both use cases are very incremental.

- The generalization to arbitrary manifolds is a strength of the paper (++).

- Experimental results are not the most impressive either. In addition, only two baselines have been compared to (e.g. in Table 1), notwithstanding the plethora of recent works on the area, including the works cited in the Related Work.

- p2: Last paragraph in the Related Work: Two things cannot be "identical", yet there are "important differences" among them.. Please fix the language.

- p2: Last paragraph in the Related Work: The differences referred to as "important differences" are very incremental and non-fundamental. Therefore, the Neural Bayes-MIM-v1 objective is not novel.

- The paper needs massive improvements in terms of writing. Ideas need to flow in order and to be much more unequivocal than the current form. Just as an example, the term E_x[L_k(x)] has been thrown in in the related work section prior to the formulation of the notation.

- p4: "we found the following finite difference Jacobian regularization to be effective.": Could you please elaborate a bit on that? Did you try other regularization procedures which turned out to be ineffective or less effective?

---

### Decision · Program_Chairs · 2021-01-07
**Final Decision**

**Decision:**

Reject

**Comment:**

Summary: The authors propose a method for representing a posterior
over discrete latent variables in representation learning problems
using a neural network. Two applications are discussed: One are certain
clustering problems, in which clusters are sufficiently
separated. Another is the computation of mutual information of
discrete random variables. This is applied to learning image representations.

Discussion: The authors have not provided a response to the reviews.

Recommendation: Four detailed reviews unanimously recommend rejection. Main
points of criticism are lack of novelty, limited and unconvincing
experimental evaluation, and a poor presentation that also lacks
technical detail. This work is clearly not ready for publication.